# Structural basis for activation of DNMT1

Amika Kikuchi [1,9], Hiroki Onoda[1,8,9], Kosuke Yamaguchi[2], Satomi Kori[1], Shun Matsuzawa[1], Yoshie Chiba [3], Shota Tanimoto[3], Sae Yoshimi[1], Hiroki Sato[1], Atsushi Yamagata [4], Mikako Shirouzu [4], Naruhiko Adachi [5], Jafar Sharif[6], Haruhiko Koseki [6,7], Atsuya Nishiyama [3], Makoto Nakanishi[3], Pierre-Antoine Defossez [2] & Kyohei Arita [1] ✉

DNMT1 is an essential enzyme that maintains genomic DNA methylation, and its function is regulated by mechanisms that are not yet fully understood. Here, we report the cryo-EM structure of human DNMT1 bound to its two natural activators: hemimethylated DNA and ubiquitinated histone H3. We find that a hitherto unstudied linker, between the RFTS and CXXC domains, plays a key role for activation. It contains a conserved α-helix which engages a crucial "Toggle" pocket, displacing a previously described inhibitory linker, and allowing the DNA Recognition Helix to spring into the active conformation. This is accompanied by large-scale reorganization of the inhibitory RFTS and CXXC domains, allowing the enzyme to gain full activity. Our results therefore provide a mechanistic basis for the activation of DNMT1, with consequences for basic research and drug design.

DNA methylation is a key epigenetic mark that regulates gene expression and genome stability[1,2]. In mammals, DNA methylation occurs at the $5^{th}$ position of cytosine, mostly within CpG dinucleotides, and it is catalyzed by a DNA methyltransferase (DNMT) family. De novo DNMTs (DNMT3A, 3B, and 3L) set up proper DNA methylation pattern during development and differentiation, and this pattern is faithfully copied on the newly replicated DNA at each round of cell division, by the maintenance enzyme DNMT1[3,4]. An E3 ubiquitin ligase (ubiquitin-like containing PHD and RING finger domains 1, UHRF1) protein, plays a crucial role for maintenance DNA methylation[5,6], together with DNMT1. UHRF1 recognizes hemimethylated DNA via its SET and RING-associated (SRA)[7–9] and catalyzes double monoubiquitination at K18 and K23 on histone H3 (H3Ub2) which, in turn, recruits DNMT1 and stimulates its enzymatic activity[10–14].

DNMT1 is a large protein (1616 amino acids), containing multiple domains (Fig. 1a), and subject to intramolecular regulations

that strongly restrict its activity to hemimethylated DNA[15]. In the absence of DNA (apo-DNMT1, PDB:4WXX, aa:351–1600), the enzyme is autoinhibited: Binding of Replication-Foci Targeting Sequence (RFTS) to the catalytic core, in association with recognition of the DNA binding region by an Auto-Inhibitory Linker, inhibit the access of hemimethylated DNA to DNMT1 catalytic region[16,17]. A key unresolved question is: how does the combined presence of H3Ub2 and hemimethylated DNA allow the enzyme to overcome this double inhibition? Of note, previous structural studies of DNMT1 in a complex with hemimethylated DNA (PDB:4DA4, aa:731–1602; 6X9I, aa:729–1600)[18–20], in a complex with unmethylated CpG DNA (PDB:3PTA, aa:646–1600)[21] and RFTS bound to H3Ub2 (PDB:5WVO, aa:351–600; 6PZV, aa:349–594)[12,13] have used a truncated version of the protein (Fig. 1a), therefore the fate of the inhibitory regions, RFTS and Auto-Inhibitory Linker, during activation is unknown.

[1]Structural Biology Laboratory, Graduate School of Medical Life Science, Yokohama City University, Tsurumi-ku, Yokohama, Kanagawa 230-0045, Japan. [2]Université Paris Cité, CNRS, Epigenetics and Cell Fate, F-75013 Paris, France. [3]Division of Cancer Cell Biology, Institute of Medical Science, The University of Tokyo, 4-6-1 Shirokanedai, Minato-ku, Tokyo 108-8639, Japan. [4]Laboratory for Protein Functional and Structural Biology, RIKEN Center for Biosystems Dynamics Research, 1-7-22 Suehiro-cho, Tsurumi-ku, Yokohama, Kanagawa 230-0045, Japan. [5]Structural Biology Research Center, Photon Factory, Institute of Materials Structure Science, High Energy Accelerator Research Organization (KEK), 1-1 Oho, Tsukuba, Ibaraki 305-0801, Japan. [6]Laboratory for Developmental Genetics, RIKEN Center for Integrative Medical Sciences (IMS), 1-7-22 Suehiro-cho, Tsurumi-ku, Yokohama, Kanagawa 230-0045, Japan. [7]Department of Cellular and Molecular Medicine, Graduate School of Medical and Pharmaceutical Sciences, Chiba University, Chiba 260-8670, Japan. [8]Present address: Synchrotron Radiation Research Center, Nagoya University, Furo-Cho, Chikusa-Ku, Nagoya 464-8603, Japan. [9]These authors contributed equally: Amika Kikuchi, Hiroki Onoda. ✉e-mail: aritak@yokohama-cu.ac.jp

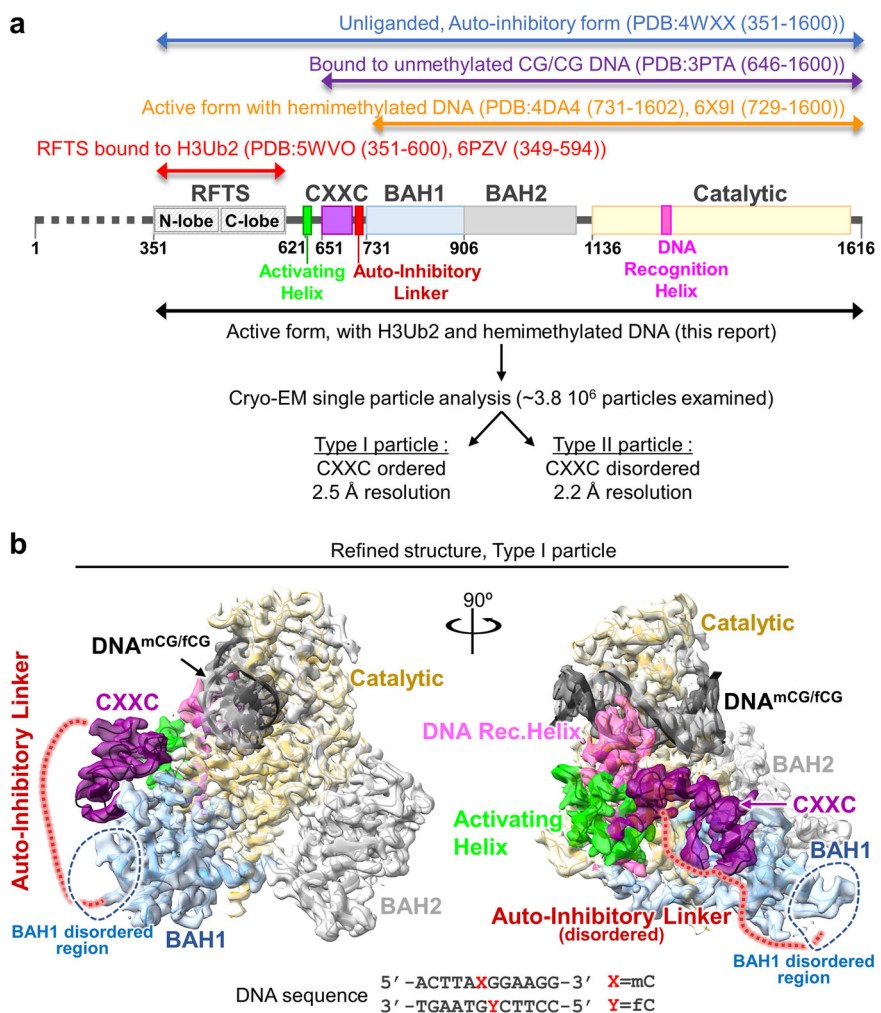

**Fig. 1 | Cryo-EM single particle analysis of DNMT1:H3Ub2:DNA^mCG/fCG. a** Domain architecture of DNMT1, with amino acid numbers indicated. **b** Cryo-EM map of the DNMT1:H3Ub2:DNA^mCG/fCG ternary complex superposed on the cartoon model. Disordered Auto-Inhibitory Linker and N-terminal portion of BAH1 domain are shown as dotted lines.

In order to understand the detailed molecular mechanism for DNMT1 activation, we have determined the cryogenic electron microscopy (cryo-EM) structure of human DNMT1 (aa:351–1616), stimulated by the H3Ub2 tail and in an intermediate complex with a hemimethylated DNA analog. Our structure illuminates the synergistic structural rearrangements that underpin the activation of DNMT1. In particular, it highlights the key role of a hydrophobic "Toggle" pocket in the catalytic domain, which stabilizes both the inactive (inhibited) or the active states. In the latter case, it functions by accepting a pair of phenylalanine residues from a hitherto unrecognized, yet highly conserved Activating Helix located between the RFTS and CXXC domains.

## Results

### Cryo-EM structure of DNMT1 bound to ubiquitinated H3 and hemimethylated DNA

To uncover the molecular mechanism of DNMT1 activation, we conducted cryo-EM single particle analysis of DNMT1 (aa:351–1616) in an intermediate complex with H3Ub2 tail and hemimethylated DNA (Fig. 1a, b). The human DNMT1 protein, a minimum fragment required for investigating the activation mechanism by binding of ubiquitinated H3, was produced using the Sf9 baculovirus expression system, and an H3 tail peptide (aa:1–37 W, K14R/K27R/K36R) was di-monoubiquitinated on K18 and K23 to completion in vitro (Supplementary Fig. 1a and see method). As expected from previous work[12,13],

in contrast to K18 or K23 single monoubiquitinated H3, the addition of H3Ub2 effectively enhanced the enzymatic activity of DNMT1 (Supplementary Fig. 1d, e). The DNMT1:H3Ub2 binary complex was used for reaction with hemimethylated DNA. The target cytosine in hemimethylated DNA was replaced by a 5-fluorocytosine (5fC) to form an irreversible covalent complex with DNMT1[22]. The ternary complex containing DNMT1 bound to H3Ub2 and DNA^mCG/fCG was purified by gel-filtration chromatography (Supplementary Fig. 2a–c), and used for cryo-EM single particle analysis (Supplementary Fig. 3a).

3D variability analysis by cryoSPARC[23] revealed 2 types of particles: Type I particles with 2.5 Å resolution, in which the CXXC domain was ordered (Fig. 1b and Supplementary Fig. 4), and Type II particles with 2.2 Å resolution, in which the CXXC was disordered (Supplementary Fig. 5a). The atomic models of DNMT1 were constructed from the two types (Supplementary Table 1); these structures were essentially the same (except for the CXXC domain), therefore in the rest of the paper we will focus on the Type I DNMT1 particle, with the ordered CXXC. In this structure, the RFTS domain, Auto-Inhibitory Linker, N-terminal β-sheet of BAH1 (aa:731–755), and some loops and linkers were invisible, reflecting their flexibility (Fig. 1b). The structured elements of the ternary complex showed that the catalytic core of DNMT1 bound to hemimethylated DNA, the catalytic loop (aa:1224–1238) recognized flipped-out 5fC, and the Target Recognition Domain (TRD) residues (Cys1499, Leu1500, Trp1510, Leu1513, Met1533, and Gly1534) bound to the methyl-group of 5mC (Supplementary Fig. 5b–d). Overall,

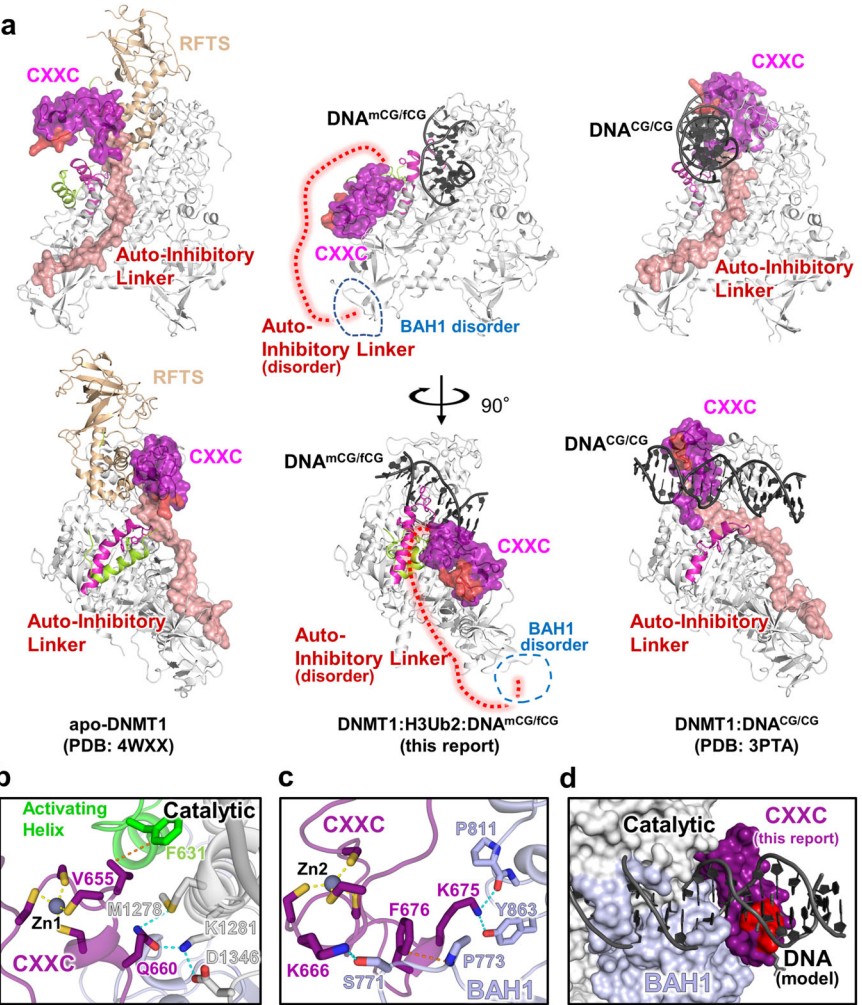

**Fig. 2 | Spatial rearrangement of the CXXC domain upon activation.**
**a** Comparison of the overall structure of apo-DNMT1 (left, PDB:4WXX), DNMT1:
H3Ub2:DNA^mCG/fCG ternary complex (center, this study), and DNMT1:DNA^CG/CG binary complex (right, PDB: 3PTA). Upper and lower panels show the side and front views of DNMT1, respectively. RFTS domain, Activating Helix, CXXC domain, Auto-Inhibitory Linker, and DNA Recognition Helix were colored light orange, light green, purple, red, and pink, respectively. CXXC domain and Auto-Inhibitory Linker are also exhibited as the transparent surface model. The disordered regions of BAH1 and Auto-Inhibitory Linker in DNMT1:H3Ub2:DNA^mCG/fCG ternary complex are shown as dotted line. The red surface in the CXXC domain indicates R681-K683, which binds the major groove of unmethylated DNA^CG/CG. **b** Detail of the interaction between the CXXC domain and catalytic domain showing gray cartoon and stick models in the ternary complex. **c** Detail of the interaction between the CXXC domain and BAH1 domain showing light-purple cartoon and stick models in the ternary complex. **d** Model structure of CXXC domain bound to DNA^CG/CG in the ternary complex. CXXC:DNA^CG/CG in PDB:3PTA is superimposed on the CXXC domain in the ternary complex. The red surface in the CXXC domain indicates the binding surface for binding to the major groove of unmethylated DNA^CG/CG.

these results confirm previously reported DNMT1:hemimethylated DNA binary complex structures (Supplementary Notes)[18–20]. In addition, they show the behavior of the RFTS and Auto-Inhibitory Linker in the active form, as described in the following section.

## Spatial rearrangement of RFTS and CXXC domains in the active form

The dissociation of the RFTS domain from the catalytic core is assumed to be required for DNMT1 activation, yet no direct evidence of this structural rearrangement has been shown to date. Our cryo-EM structure of apo-DNMT1 (aa:351–1616) at 3.4 Å resolution showed a fully-structured RFTS domain bound to the catalytic core (Supplementary Fig. 6a left, and Supplementary Table 2). In contrast, the cryo-EM map of the DNMT1:H3Ub2:DNA^mCG/fCG ternary complex showed no density for the RFTS:H3Ub2 (Fig. 1b). Intriguingly, particles smaller than the DNMT1:H3Ub2:DNA^mCG/fCG ternary complex were observed in 2D class average (Supplementary Fig. 7a). The size and shape of these particles were comparable to those of RFTS:H3Ub2 as estimated from the 2D projected template of RFTS:H3Ub2 (PDB:5WVO)-derived 3D

Gaussian model (Supplementary Fig. 7b). These data indicate that the RFTS:H3Ub2 moiety is in a highly dynamic state and does not interact with other domains when DNMT1 is active.

To separate the contributions of H3Ub2 and of hemimethylated DNA for displacement of the RFTS, we determined the cryo-EM structure of DNMT1 (aa:351–1616) in complex with H3Ub2, but without DNA (Supplementary Figs. 1b, 6a right, and Supplementary Table 2). This structure reached 3.6 Å resolution and showed that, while the N-lobe of DNMT1 became flexible and therefore invisible, the C-lobe remained bound firmly to the catalytic core upon H3Ub2 binding. These data were further supported by small angle X-ray scattering (SAXS) analyses (Supplementary Figs. 6b, 8a–c, Supplementary Table 3, and Supplementary Notes), showing that H3Ub2, in itself, is not sufficient to dislodge the RFTS.

We next investigated the dynamics of the CXXC domain after DNMT1 activation. In apo-DNMT1, the CXXC domain is affixed to the side of the RFTS domain (Fig. 2a left)[16,17]. In the presence of unmethylated DNA (DNA^CG/CG) (PDB:3PTA), the zinc finger motif of the CXXC domain recognizes unmethylated CpG, shifts 30 Å towards the TRD

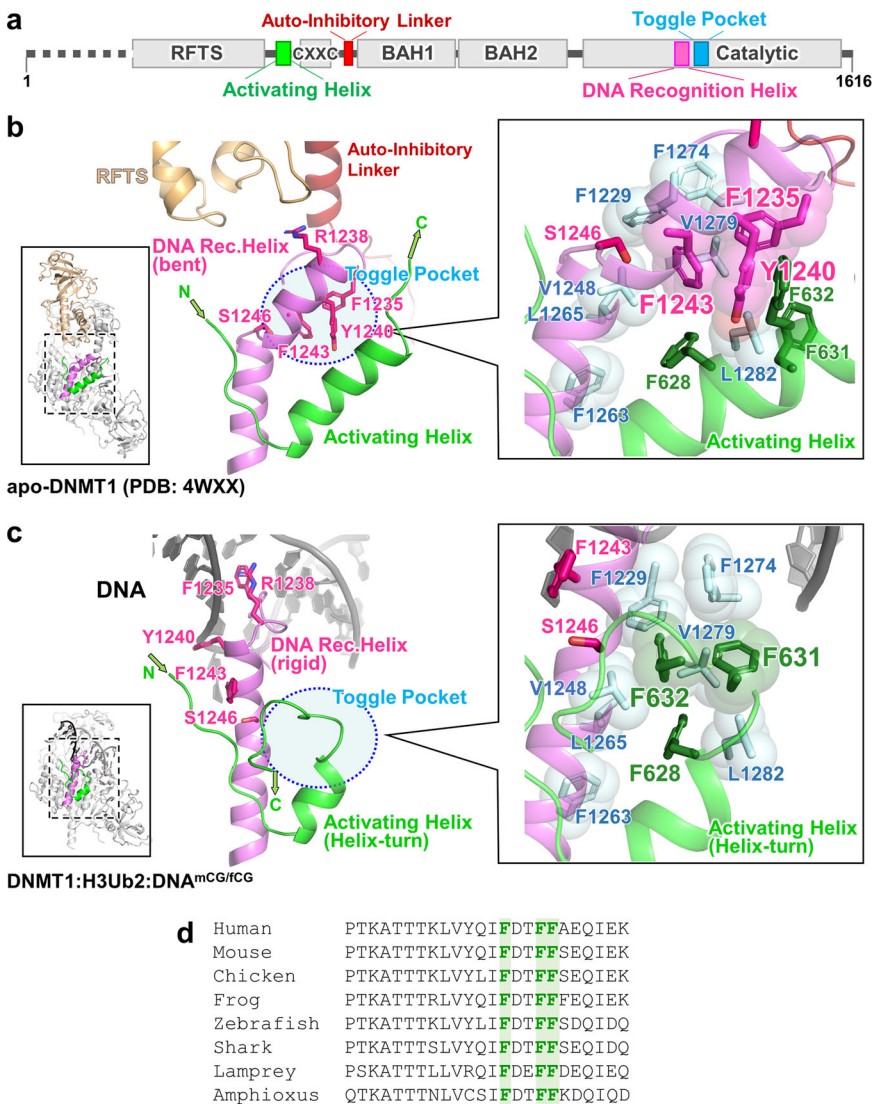

**Fig. 3 | Activation of DNMT1 by a pair of phenylalanine residues in the Activating Helix. a** Domain structure of DNMT1. **b** Structure around Activating Helix (green), DNA Recognition Helix (pink), and Toggle Pocket (pale cyan) of apo-DNMT1 (PDB:4WXX). The Toggle Pocket is highlighted by a light blue circle. The right panel is a magnified figure of the Toggle Pocket. Residues in Activating Helix and DNA Recognition Helix that are involved in binding to the Toggle Pocket are shown as green and pink stick models, respectively. Residues for the formation of the Toggle Pocket are shown as a stick with a transparent sphere model. **c** Structure around Activating Helix (green), DNA Recognition Helix (pink), and Toggle Pocket (pale cyan) of the ternary complex. The color scheme is the same as Fig. 3b. **d** Multiple sequence alignment of DNMT1 around the Activating Helix. Phenylalanines 628, 631, and 632 are highlighted.

domain, and sits near the active center (Fig. 2a right)[21]. In this conformation, the Auto-Inhibitory Linker directly interrupts the binding of DNA to the active site, which prevents unlicensed de novo DNA methylation. Intriguingly, in our active complex, the CXXC domain moved away from the active center and took an "upside-down" position (relative to 3PTA), between the BAH1 domain and the catalytic domain with which it established hydrogen bonds and van der Waals interactions (Fig. 2b, c).

As the DNA binding interface of the CXXC remained solvent-exposed in the active complex (Fig. 2a center), so we asked whether it could still bind unmethylated DNA. However, the superimposition of the CXXC:DNA complex onto the ternary complex structure revealed major steric clashes with the catalytic domain (Fig. 2d), suggesting that DNA binding by the CXXC is fully suppressed when DNMT1 is active. The new position of the CXXC leads to a structural deformation of the BAH1 N-terminal β-sheet and induces a different orientation of the BAH1 loop (aa:765–775) (Figs. 1b, 2a center). These structural changes then lead to full eviction of the Auto-Inhibitory Linker from the

catalytic core and causes the Auto-Inhibitory Linker to adopt a highly disordered structure (Figs. 1b, 2a).

Taken together, our cryo-EM analysis of the active form of DNMT1 revealed a large reorganization of the inhibitory domains relative to unliganded or inactive forms. Upon joint binding of H3Ub2 and hemimethylated DNA, the RFTS domain is forced out of the catalytic core, while the CXXC domain undergoes a drastic spatial rearrangement, ultimately leading to the eviction of the Auto-Inhibitory Linker from the catalytic core.

## A "Toggle" pocket accepts different phenylalanines in the repressed and active states

Zooming in on the catalytic site, we observed another striking difference between the inactive and active states of the enzyme.

In the inactive state, the DNA Recognition Helix (aa:1236–1259) in the catalytic domain is kinked, at Ser1246 (Fig. 3b). A hydrophobic pocket (composed of Phe1229, Val1248, Phe1263, Leu1265, Phe1274, Val1279, and Leu1282, hereafter Toggle Pocket) accepts Phe1235 of the

catalytic loop and Phe1243 of the DNA Recognition Helix (Fig. 3a). In addition, Tyr1240 within this same helix forms hydrophobic interactions with Phe628, Phe631, and Phe632 from the Activating Helix (aa:621–635), further stabilizing the inactive conformation (Fig. 3b).

This contrasts sharply with our activated form of DNMT1 (Fig. 3c). In that situation, the Activating Helix is shortened as it forms a helix-turn; its residues Phe631 and Phe632 invade the Toggle Pocket. The DNA Recognition Helix is freed from the Toggle Pocket and springs into a straight conformation, which allows (i) access of Phe1235/Arg1238 (in the catalytic loop) to the minor groove at the mCG/fCG site, (ii) formation of a hydrogen bond between Tyr1240 in the DNA Recognition Helix and the phosphate backbone of the DNA, and (iii) engagement of Phe1243 (in the DNA Recognition Helix) by Pro615, Lys617, and Gln635 (in the linker between RFTS and CXXC domains), which prevents Phe1243 from entering the Toggle Pocket (Fig. 3a, c and Supplementary Fig. 9a).

Interestingly, a previous study of human DNMT1 binary complex (aa:729–1600, not containing Activating Helix) with hemimethylated DNA analog (active state, PDB: 6X9I) shows that Phe1243 remains in the Toggle Pocket and the N-terminal region of the DNA Recognition Helix structure is partially unfolded, thereby preventing the binding of Tyr1240 to the phosphate backbone of DNA (Supplementary Fig. 9a, b)[20]. Furthermore, folded or unfolded DNA Recognition Helix structures are observed in the previous structures of mouse DNMT1 (aa:731–1602) bound to hemimethylated DNA analog, depending on the sequence around the mCG/fCG[18,19], suggesting that the DNA Recognition Helix is intrinsically flexible; in contrast, our DNMT1:H3Ub2:DNA$^{mCG/fCG}$ complex shows a rigid conformation of the DNA Recognition Helix, indicating that the phenylalanine pair in the Activating Helix contributes to the activation state of DNMT1.

### Crucial role of a phenylalanine pair for activation of DNMT1

The phenylalanines Phe631 and Phe632 are invariant between vertebrate species, and are also present in the cephalochordate Amphioxus (Fig. 3d). We, therefore, asked whether these residues played a role in the activation of DNMT1, as could be expected from the fact that they bind the Toggle Pocket. In a binding assay, we found that the F631A/F632A mutations abolished the ability of DNMT1 to bind hemimethylated DNA (Fig. 4a), even though the H3Ub2-binding ability of DNMT1 was unaffected (Supplementary Fig. 1f). We carried out an in vitro DNA methylation assay and, again, observed that the F631A/F632A mutation led to severe defects in DNA methylation (Fig. 4b).

We then sought confirmation of these results in an in vitro assay, which reconstitutes replication-coupled maintenance DNA methylation using Xenopus egg extracts[11,12,24]. In that system, we immunodepleted DNMT1, and re-introduced recombinant DNMT1, either WT or mutated on the two phenylalanines of the Activating Helix (F506A and F507A in Xenopus, FF/AA mutant). As previously reported, the depletion of xDNMT1 from Xenopus egg extracts resulted in the accumulation of chromatin-bound UHRF1 and ubiquitinated histone H3 species (Fig. 4c); this is due to defective maintenance DNA methylation, which generates hemimethylated DNA from which UHRF1 cannot be released[11,24]. The addition of wild-type (WT) recombinant xDNMT1 suppressed the accumulation of UHRF1 and ubiquitinated H3 (Fig. 4c). The FF/AA mutant retained chromatin binding activity but failed to suppress the accumulation of UHRF1 and ubiquitinated H3, showing defects in maintenance DNA methylation (Fig. 4c). Therefore, this functional assay in Xenopus egg extracts further validated the effect of the mutation.

Lastly, we used a colon cancer cell line HCT116, in which DNA methylation has been widely studied. In this line, both endogenous alleles of DNMT1 are tagged with an auxin-inducible degron (AID) (Fig. 4d). In this DNMT1-AID line, we introduced rescue vectors: one encoding WT DNMT1, and the other encoding the FF/AA mutant (Fig. 4d). The level of endogenous DNMT1, exogenous WT or FF/AA DNMT1 were comparable (Fig. 4e), and they were located in the

nucleus (Supplementary Fig. 10). Treating the cells with indole-3-acetic acid (IAA) caused the disappearance of endogenous DNMT1 but did not affect the exogenous proteins. We then measured global DNA methylation at days 0, 4, and 8 after endogenous DNMT1 removal (Fig. 4f). The control cells (empty vector) lost almost one-third of total DNA methylation, going from 70 to 50% methylation. This loss was completely prevented by the WT DNMT1 transgene. In contrast, the FF/AA DNMT1 mutant was incapable of sustaining DNA methylation maintenance and showed DNA methylation values close to those of the empty vector (Fig. 4f).

Collectively these experiments confirm that the Activating Helix, and especially its conserved phenylalanines, are crucial for DNA methylation maintenance by DNMT1.

## Discussion

Our cryo-EM analysis reveals a molecular mechanism for human DNMT1 catalytic activation (Fig. 5). Our results reveal both the large-scale displacements of inhibitory modules (RFTS, CXXC, Auto-Inhibitory Linker), as well as more detailed changes, particularly the switch by which the same hydrophobic pocket, initially bound to inhibitory phenylalanines, engages activating phenylalanines, which releases the DNA Recognition Helix and permits catalysis. This regulation also operates in Xenopus, and may even occur in invertebrates such as Amphioxus, in which the regulatory amino acids are conserved (Fig. 3d).

The catalytic domains of the de novo DNA methyltransferases DNMT3A and DNMT3B bind the DNMT3L catalytic-like domain and form a heterotetramer[25–28]. Interestingly, the DNMT3A(B)/3 L interface is formed by hydrophobic interactions mediated by phenylalanine residues, and therefore is known as the F-F interface. The F-F interface enhances DNA methylation activity by the DNMT3A(B)/3L heterotetramer[25]. The hydrophobic residues in the DNMT3A(B) catalytic domain spatially corresponds to the Toggle Pocket of DNMT1. Thus, covering the hydrophobic pocket of the catalytic domain by an intra- or inter-molecule interaction could be an evolutionarily conserved activation mechanism of DNA methyltransferases. The Activating Helix, however, is unique to DNMT1 and crucial for enzymatic activation, and therefore could be utilized to design novel inhibitors such as helical peptides that mimic this Activating Helix.

Previously reported structures of apo-DNMT1 and DNMT1:DNA$^{CG/CG}$ revealed a dual-auto-inhibitory mechanism in which the RFTS domain and the Auto-Inhibitory Linker are embedded into the catalytic core, thereby inhibiting the access of cognate DNA (Fig. 5). Our cryo-EM analysis of the ternary complex showed full dissociation of both the RFTS domain and Auto-Inhibitory Linker from the catalytic core (Fig. 1b). Interestingly, H3Ub2-binding to the RFTS domain might not be sufficient for the displacement of the RFTS domain as our cryo-EM and SAXS data showed that the C-lobe of RFTS domain is still accommodated in the catalytic core in the DNMT1:H3Ub2 complex (Supplementary Fig. 6). A previous molecular dynamics simulation has demonstrated that H3Ub2-binding reduces the contact number between the C-lobe and catalytic core[12]. In addition, apo-DNMT1 was unable to form the binary complex with hemimethylated DNA (Supplementary Fig. 1c). We hypothesize, therefore, that H3Ub2 binding destabilizes the inhibitory interaction between the C-lobe and the catalytic core, allowing hemimethylated DNA to penetrate the catalytic core. Thus, we propose that simultaneous binding to H3Ub2 and DNA leads to full activation of DNMT1 via the following structural changes: (i) dissociation of the RFTS domain from the catalytic core, (ii) structural changes to an Activating Helix causing the conserved residues Phe631 and Phe632 to invade the Toggle Pocket of the catalytic domain, (iii) adoption of a rigid conformation by the DNA Recognition Helix, (iv) spatial rearrangement of the CXXC domain, and (v) eviction of the Auto-Inhibitory Linker from the catalytic domain (Fig. 5). However, it is currently unknown how simultaneous binding of H3Ub2 and hemimethylated DNA causes a conformational change in the Activating Helix

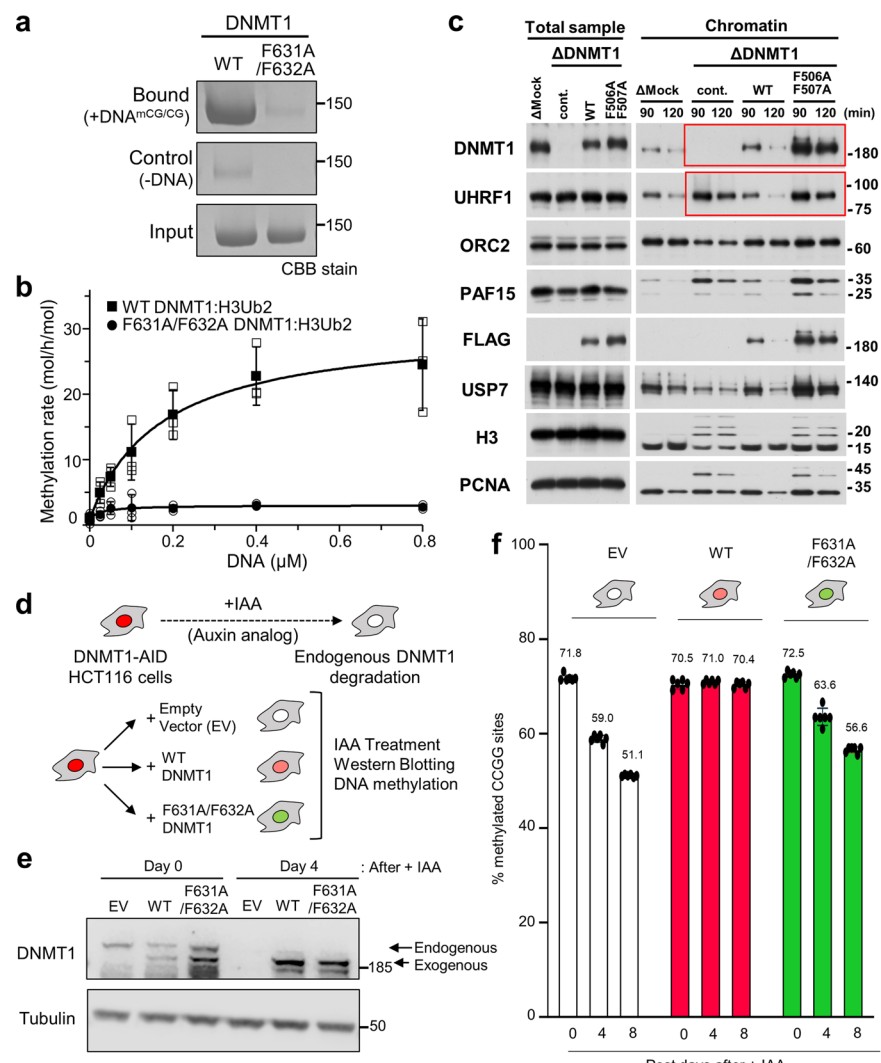

**Fig. 4 | Functional assays demonstrate the importance of the Activating Helix.**
**a** Pull-down assay using immobilized hemimethylated DNA. Three independent
experiments were performed. **b** in vitro DNA methylation activity of DNMT1 WT
(black square: mean value, white square: three independent experiment values) and
F631A/F632A mutant (blacks circle: mean value, white circle: three independent
experiment values) for hemimethylated DNA. The vertical axis indicates the turn-
over frequency of the methylation reaction of DNMT1 (15 nM) after 1 h.
Michaelis–Menten curve is shown as lines. Data were presented as mean values ± SD
for $n = 3$. **c** Interphase extracts depleted with either control or xDNMT1 antibodies
were incubated with control buffer, purified recombinant WT xDNMT1, and F506A/
F507A. The chromatin fractions were isolated at the indicated times and their
bound proteins, as well as inputs, were analyzed by immunoblotting. The gel image
is representative of $n = 3$ independent experiments. **d** Experimental scheme in
human HCT116 colon cancer cells: the endogenous DNMT1 protein is tagged with
an auxin-inducible degron (AID) tag, causing the protein to be degraded after the
addition of the auxin analog IAA. In this background, rescue vectors are added that
encode either WT or FF/AA versions of DNMT1. The empty vector is used as a
negative control. **e** Western blotting shows that endogenous DNMT1 is degraded
upon IAA addition, whereas exogenous WT and FF/AA DNMT1 are not degraded.
The repetitive of the experiment is $n = 1$. **f** Measurement of global DNA levels by
luminometric methylation assay (LUMA). Cells expressing WT DNMT1 maintain
global DNA methylation when endogenous DNMT1 is degraded, whereas the FF/AA
mutant does not support DNA methylation maintenance. Data represent the
mean ± SD of $n = 3$ independent experiments. Source data of Figs. 4a–c, 4e, f are
provided as a Source Data file.

to place Phe631 and Phe632 in the Toggle Pocket. Future work, such as
molecular dynamics simulation, will determine if these structural
changes occur sequentially or simultaneously. Thus, our findings
describe concepts and mechanisms in the multi-step activation process
of DNMT1 that ensures faithful maintenance of DNA methylation.

## Methods
### Oligonucleotides
Twelve bases of oligonucleotides (upper: 5′-ACTTA(5mC)GGAAGG,
lower: 5′-CCTTC(5fC)GTAAGT) for cryo-EM single particle analysis, 42
bases of oligonucleotides (upper: 5′-GGACATC(5mC)GTGAGATCGGA
GGC(5mC)GCCTGCTGCAATC(5mC)GGTAG, lower: 5′-CTACCGGATTG
CAGCAGGCGGCCTCCGATCTCACGGATGTCC) for DNA methylation

assay and 21 bases of oligonucleotides (upper: 5′-CAGGCAATC(5mC)
GGTAGATCGCA, lower: 5′-biotin-TTGCGATCTACCGGATTGCCTG) for
DNA pull-down assay were synthesized by GeneDesign, *Inc.* (Osaka,
Japan). 5mC and 5fC mean 5-methylcytosine and 5-fluorocytosine,
respectively. To prepare the DNA duplex, the mixture of equimolar
complementary oligonucleotides were heated at 95 °C for 2 min and
annealed at 4 °C for overnight.

### Protein expression and purification
The gene encoding wild type and mutant of human DNMT1 (residues
351–1616) containing N-terminal ten histidine tag (His-tag) and human
rhinovirus 3 C (HRV 3 C) protease site was amplified by PCR and cloned
into the pFastBac vector (Invitrogen) using the seamless cloning

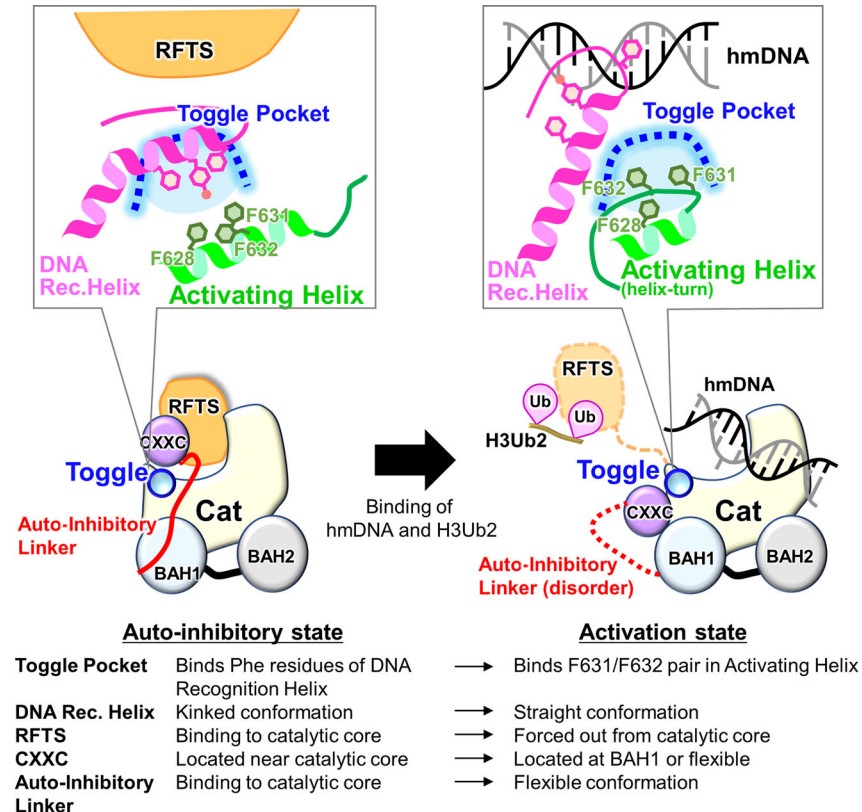

**Fig. 5 | Schematic representation of the DNMT1 activation mechanism.** Left and right figures show the auto-inhibitory and activation states of DNMT1, respectively.

method. Baculoviruses for expression of the DNMT1 were generated in *Spodoptera frugiperda* 9 (Sf9) cells according to the Bac-to-Bac system instruction (Invitrogen). The protein expression was performed by infection of the baculoviruses with the Sf9 cells for 72 hrs at 27 °C. Cells cultured with 1 L medium were lysed by lysis buffer (50 mM Tris-HCl [pH 8.0] containing 500 mM NaCl, 25 mM imidazole, 10% Glycerol, and 1 mM DTT) and sonicated with the cycle of pulse on for 10 s and pulse off for 50 s (total pulse on time: 6 min). A soluble fraction was obtained after centrifugation of the lysate at 43,667×g for 40 min at 4 °C performed using Avanti J-E with a rotor JA-20 (BECKMAN COULTER) to remove insoluble debris.

The His-tagged DNMT1 was loaded to Ni-Sepharose 6 Fast Flow (Cytiva), and unbound proteins were washed by wash buffer (50 mM Tris-HCl [pH 8.0] containing 1 M NaCl, 25 mM imidazole, 10% Glycerol and 1 mM DTT), and lysis buffer. Bound proteins were eluted by an elution buffer (20 mM Tris-HCl [pH 8.0] containing 500 mM imidazole, 300 mM NaCl, 10% Glycerol, and 1 mM DTT). The His-tag was cleaved by the HRV3C protease at 4 °C for over 12 hrs. The DNMT1 was separated by anion-exchange chromatography, HiTrap Q HP (Cytiva) using gradient elution from 50 to 1000 mM NaCl in 20 mM Tris-HCl [pH 8.0] buffer containing 10% Glycerol and 1 mM DTT. At the final stage, the DNMT1 was purified with Hiload 26/600 Superdex 200 size exclusion chromatography (Cytiva) equilibrated with 20 mM Tris-HCl [pH 7.5], 250 mM NaCl, 10% Glycerol, and 5 mM DTT.

F631A/F632A site-directed mutagenesis was performed by designing two primers containing the mutations (Forward: 5′-CGA-TACTGCTGCTGCAGAGCAAATTGAAAAGGA-3′, Reverse: 5′-GCTCTGC AGCAGCAGTATCGAAGATCTGGTAGA-3′).

**Preparation of disulfide- and isopeptide-linked ubiquitinated H3**
Disulfide-linked ubiquitinated H3 analog for SEC-SAXS and in vitro DNA methylation assay was prepared using G76C mutant of ubiquitin (Ub-G76C) and K18C/K23C mutant of H3 peptide (residues

1–36 with an additional tryptophan residue at their C-terminus, hereafter H3$_{1-37W}$-K18C/K23C). Expression in *E. coli* and purification of the Ub-G76C and H3$_{1-37W}$ were purified according to the previous report[12]. After purification, these proteins were lyophilized. The lyophilized Ub-G76C was dissolved in 50 mM sodium phosphate (pH 7.5) and mixed with a 20-fold molar excess of 5,5′-dithiobis-(2-nitrobenzoic acid) (DTNB, Wako) and the mixture was incubated for 40 min at room temperature with rotation on a ROTATOR RT-5 (TITEC). The reaction solution was buffer-exchanged into ligation buffer (20 mM Tris-HCl [pH 7.0], 50 mM NaCl, and 1 mM EDTA) using a PD-10 desalting column (Cytiva). Lyophilized H3$_{1-37W}$-K18C/K23C was reduced in 20 mM Tris-HCl (H 7.5) containing 5 mM DTT, and were buffer-exchanged into ligation buffer, and mixed with a 5-fold molar excess of activated Ub-G76C-DTNB for 1 hr. The reaction product was purified on a cation-exchange column, Mono-S (Cytiva).

Isopeptide-linked ubiquitinated H3 for cryo-EM single particle analysis was prepared using mouse UBA1 (E1), human UBE2D3 (E2), human UHRF1 (E3), ubiquitin and H3$_{1-37W}$ harboring K14R/K27R/K36R mutations (H3$_{1-37W}$ K14R/K27R/K36R), which were purified in house as previous report[24]. The ubiquitination reaction mixture contained 0.2 μM E1, 0.8 μM E2, 3 μM E3, 150 μM ubiquitin, and 50 μM H3$_{1-37W}$ K14R/K27R/K36R mutant in 1 mL of ubiquitination reaction buffer (50 mM Tris-HCl [pH 8.5], 50 mM NaCl, 5 mM MgCl$_2$, 0.1% Triton X-100, and 2 mM DTT). The reaction mixture was incubated at 30 °C for 6 hrs, and thereafter the reaction was quenched by the heat shock at 70 °C for 30 min, followed by collecting soluble fraction (Supplementary Fig. 1a). About 6 μM of 18 bp hemimethylated DNA was added in initially ubiquitination reaction to enhance the ubiquitination reaction, however, the DNA addition was omitted for the sample preparation of DNMT1:H3Ub2:DNA$^{mCG/fCG}$ to prevent DNA contamination.

## Preparation of binary and ternary complex for cryo-EM single particle analysis

For the preparation of DNMT1:H3Ub2 binary complex, 20 μM isopeptide bound-ubiquitinated H3$_{1-37}$ mixed with 6.6 μM apo-DNMT1 was subjected to size excursion chromatography (Superdex® 200 Increase 10/300 GL, Cytiva) equilibrated with the buffer (20 mM Tris-HCl [pH 7.5], 250 mM NaCl, 10 μM Zn(OAc)$_2$, and 0.5 mM DTT).

For the preparation of ternary complex, purified DNMT1:H3Ub2 was mixed with 12 base pair of hemimethylated DNA analog (upper: 5′-ACTTA(5mC)GGAAGG, lower: 5′-CCTTC(5fC)GTAAGT: DNA$^{mCG/fCG}$) in the conjugation buffer (50 mM Tris-HCl [pH 7.5], 20% Glycerol, 5 mM DTT, and 50 mM NaCl). The conjugation reaction was initiated by the adding of 500 μM S-adenosyl-L-methionine (SAM) at 25 °C for 4 hrs. The yield was purified by size exclusion chromatography (Superdex® 200 Increase 10/300 GL, Cytiva) equilibrated with cryo-EM buffer (20 mM Tris-HCl [pH 7.5], 250 mM NaCl, and 5 mM DTT).

## Cryo-EM data collection

A 3 μL of the protein solutions was applied onto the glow-discharged holey carbon grids (Quantifoil Cu R1.2/1.3, 300 mesh). The grids were plunge-frozen in liquid ethane using a Vitrobot Mark IV (Thermo Fisher Scientific). Parameters for plunge-freezing were set as follows: blotting time, 3 s; waiting time, 0 sec; blotting force, 0; humidity, 100%; and chamber temperature, 4 °C. Data for DNMT1:H3Ub2:DNA$^{mCG/fCG}$ ternary complex was collected at the University of Tokyo on a 300 kV Titan Krios electron microscope (Thermo Fisher Scientific) with a K3 direct electron detector (Gatan) with BioQuantum energy filter in counting mode. A total of 4068 movies were recorded at a nominal magnification of ×105,000 and a pixel size of 0.83 Å/pixel, with a defocus range between −0.8 and −1.8 μm and a dose rate of 1.25 electrons/Å$^2$ per frame. The data were automatically acquired using the SerialEM 3.9.0 software[29]. A typical motion-corrected cryo-EM image is shown in Supplementary Fig. 11a.

Data of apo-DNMT1, DNMT1:H3Ub2 were collected at RIKEN BDR on a 200-kV Tecnai Arctica electron microscope (Thermo Fisher Scientific) with a K2 direct electron detector (Gatan) in counting mode. A total of 2071 movies for apo-DNMT1 and 1869 movies for DNMT1:H3Ub2 were recorded at a nominal magnification of ×23,500 and a pixel size of 1.477 Å/pixel, with a defocus range between −0.8 and −1.4 μm, and a dose rate of 1.25 electrons/Å$^2$ per frame. A typical motion-corrected cryo-EM image is shown in Supplementary Figs. 11b, c. The data were automatically acquired using the SerialEM 3.8 software.

## Data processing

All data were processed using cryoSPARC v3.2.0[30] for PDB deposition. The movie stacks were motion corrected by Full-frame motion correction or Patch motion correction. The defocus values were estimated from Contrast transfer function (CTF) by Patch CTF estimation or CTFFIND4[31]. A total of 4,307,107 particles of DNMT1:H3Ub2:DNA$^{mCG/fCG}$ ternary complex were automatically picked using a blob picker with 80, 105, and 130 Å circular blobs and 80–130 Å elliptical blobs (Supplementary Fig. 11a). Particles (3,798,046) were then extracted in a box size of 256 pixels with a 0.83 Å/pixel size followed by a single round of reference-free 2D classifications (Supplementary Fig. 11a). The selected good class containing 1,621,988 particles were used for ab initio 3D reconstruction (Supplementary Fig. 3a). Then, non-uniform refinement was performed against all the extracted particles to yield the cryo-EM map with an overall resolution of 2.09 Å resolution. The subsequent heterogeneous refinement selected 2,653,627 particles as a good class. These particles were subjected to a 3D variability analysis, separating the CXXC-ordered and CXXC-disordered models. The particles (138,662) in the CXXC-ordered model and those (897,446) in the CXXC-disordered models were then subjected to non-uniform refinement to generate a cryo-EM map with an overall resolution of 2.52 and 2.23 Å, respectively. The classification processes were shown in

Supplementary Fig. 3a, and the statics of data collection and refinement, and validation were shown in Supplementary Table 1.

A total of 3,984,637 particles of apo-DNMT1 were automatically picked using a blob picker with 80, 105, and 130 Å circular blobs (Supplementary Fig. 11b). Particles (3,984,637) were then extracted in a box size of 160 pixels with a 1.477 Å/pixel size, followed by two rounds of reference-free 2D classifications to remove junk particles (Supplementary Fig. 11b). The selected 3,666,067 particles were subjected for ab initio 3D model reconstruction to generate four cryo-EM maps (Supplementary Fig. 3c). An initial model shows similar shape with the crystal structure of DNMT1 (PDB: 4WXX). Then, non-uniform refinement was performed against the particles classified in the initial model of DNMT1 (1,824,727). These particles were re-extracted in a box size of 256 pixels with a 1.477 Å/pixel size by local motion correction. Non-uniform refinement yields the cryo-EM map with an overall resolution of 3.32 Å resolution. These particles were subjected to a 3D variability analysis and heterogeneous refinement, removing the dimer particles, and separating the RFTS-free map and RFTS-bound map. To improve the cryo-EM map, further 3D variability analysis and clustering by PCA was performed. The particles (380,989) were then subjected to non-uniform refinement to yield the final cryo-EM map. The overall resolution of 3.45 Å resolution using the gold-standard Fourier shell correlation with a 0.143 cut-off. The classification processes were shown in Supplementary Fig. 3c, and the statics of data collection and refinement and validation were shown in Supplementary Table 2.

A total of 2,463,410 particles of DNMT1:H3Ub2 were automatically picked using a blob picker with 80, 105, and 130 Å circular blobs (Supplementary Fig. 11c). The particles were then extracted in a box size of 160 pixels in a 1.477 Å/pixel size using cryoSPARC followed by initial dataset cleanup using reference-free 2D classifications (Supplementary Fig. 11c). The selected 2,336,267 particles were subjected for ab initio 3D model reconstruction to generate four cryo-EM maps (Supplementary Fig. 3b). Further 2D classifications were performed by the 2 class of ab initio 3D model assigned as DNMT1 particles. The particles (160,088) belonging to the best four 2D classes were subjected to create the fine initial model. For further refinement, the particles (1,303,645) without ice images from 2D classification II were selected. These particles were re-extracted in a box size of 256 pixels with a 1.477 Å/pixel size by local motion correction. Non-uniform refinement yields the cryo-EM map with an overall resolution of 3.55 Å resolution. These particles were subjected to a 3D variability analysis and heterogeneous refinement, separating the RFTS-free map and the RFTS-bound map. The subsequent 2D classification selected 735,233 particles as RFTS-bound structures. To improve the cryo-EM map, further 3D variability analysis and clustering by PCA was performed. The particles (645,368) were then subjected to non-uniform refinement to yield the final cryo-EM map. The overall resolution of 3.52 Å resolution using the gold-standard Fourier shell correlation with a 0.143 cut-off. The classification processes were shown in Supplementary Fig. 3b, and the statics of data collection and refinement and validation were shown in Supplementary Table 2.

For the analysis of the H3Ub2-RFTS domain complex, the 2D classification analysis were also performed by RELION-3.1 (Supplementary Fig. 7)[32]. The movie stacks of DNMT1:H3Ub2:DNA$^{mCG/fCG}$ were motion corrected by MotionCor2. The defocus values were estimated from CTF by CTFFIND4[31]. Single particle image was also extracted by LoG Auto picker of RELION to check the particles of other biomolecules. After three rounds of 2D classification, the particle smaller than DNMT1 was selected. These smaller particles (80,186) were re-extracted in a box size of 128 pixels with a 0.83 Å/pixel size, and the images were classified by 2D classification. The major 2D average images were compared with the projected templates of the RFTS-H3Ub2 complex (PDB: 5WVO) (Supplementary Fig. 7). Gaussian model of the complex was created by the Molmap of ChimeraX[33]. The 2D projected templates were created by the module "create template" in cryoSPARC.

## SEC-SAXS

SAXS data were collected on Photon Factory BL-10C using an HPLC Nexera/Prominence-I (Shimazu) integrated SAXS set-up. About 100 µL of 10 mg/mL of the apo-DNMT1 (aa:351–1616) or its bound to H3Ub2$^{S-S}$ were loaded onto a Superdex® 200 Increase 10/300 GL (Cytiva) pre-equilibrated with 20 mM Tris·HCl (pH 8.0), 150 mM NaCl and 5% glycerol at a flow rate of 0.5 mL/min at 20 °C. The flow rate was reduced to 0.05 mL/min at an elution volume of 10–13 mL. X-ray scattering was collected every 20 s on a PILATUS3 2 M detector over an angular range of $q_{min} = 0.00690$ Å$^{-1}$ to $q_{max} = 0.27815$ Å$^{-1}$. UV spectra at a range of 200 to 450 nm were recorded every 10 s. Circular averaging and buffer subtraction were carried out using the program SAngler[34] to obtain one-dimensional scattering data $I(q)$ as a function of $q$ ($q = 4\pi\sin\theta/\lambda$, where $2\theta$ is the scattering angle and $\lambda$ is the X-ray wavelength 1.5 Å). The scattering intensity was normalized on an absolute scale using the scattering intensity of water[35]. The multiple concentrations of the scattering data around the peak at A280, namely ascending and descending parts of the chromatography peak, and $I(0)$ were extrapolated to zero-concentration by Serial Analyzer[36]. The molecular mass of the measured proteins was estimated by the empirical volume of correlation, $V_c$, showing no aggregation of the measured sample[37]. The radius of gyration $R_g$ and the forward scattering intensity $I(0)$ were estimated from the Guinier plot of $I(q)$ in the smaller angle region of $qR_g < 1.3$. The distance distribution function $P(r)$ was calculated using the program GNOM[38]. The maximum particle dimension $D_{max}$ was estimated from the $P(r)$ function as the distance $r$ for which $P(r) = 0$. The scattering profile of the crystal structure of apo-DNMT1 and its docking model with ubiquitinated H3 were computed with CRYSOL[39].

## In vitro DNA methylation assay

The 42 base pair of DNA duplex containing three hemimethylation sites (0–0.8 µM) was methylated with the recombinant DNMT1 (15 nM, aa:351–1616) by the addition of the disulfide-linked ubiquitinated H3 (1 µM H3Ub2$^{S-S}$) including 20 µM SAM in reaction buffer (20 mM Tris·HCl [pH 8.0], 50 mM NaCl, 1 mM EDTA, 3 mM MgCl$_2$, 0.1 mg/mL BSA, and 20% Glycerol) at 37 °C for 1 hr. Termination of methylation reaction and conversion of SAH to ADP were performed by the addition of 5×MTase-Glo$^{TM}$ reagent from methyltransferase assay kit, MTase-Glo (Promega) at 1:4 ratio for the reaction total volume. After 30 min stationary at room temperature, the ADP detection process was carried out with solid white flat-bottom 96-well plates (Costar). MTase-Glo$^{TM}$ Detection Solution was added to the reaction in a 1:1 ratio to reaction total 40 µL volume and incubated for 30 min at room temperature. The luminescence derived from the reaction product, SAH, was monitored using GloMax® Navigator Microplate Luminometer (Promega). The effect of F631A/F632A mutation were examined at the condition of the DNMT1 (15 nM) with 1 µM H3Ub2$^{S-S}$ by the addition of 42 base pair of DNA duplex (0–0.8 µM) in the same reaction buffer. The SAH conversion process and ADP detection process are in the manner described above.

For the evaluation of the DNMT1:H3Ub2 complex, the final concentration of DNMT1 was 50 nM to prevent the dissociation of ubiquitinated H3 ($K_D = 18$ nM). DNA methylation reactions were initiated by mixing of apo-DNMT1 or DNMT1:H3Ub2$^{iso}$ and stopped at 0, 5, 15, or 30 min by addition of 5×MTase-Glo$^{TM}$ reagent. The detection process was performed in the same way as described above. At least three independent experiments were performed for the estimation of standard deviation.

## DNA pull-down assay

About 20 µg of the 21-base pair of biotinylated hemimethylated DNA duplex was immobilized on Dynabeads M-280 Streptavidin (VERITAS) equilibrated with the binding buffer (20 mM Tris·HCl [pH 7.5], 150 mM NaCl, 10% Glycerol, and 0.05% Nonidet P-40 (NP-40)). After washing the beads with the binding buffer, 10 µg of purified DNMT1 (aa:351–1616) wild-type or F631A/F632A mutant, 2-equimolar excess of

H3Ub2$^{S-S}$ and equimolar of SAH were added to the beads. After incubation for 2 hrs at 4 °C, the unbound proteins were washed five times with the binding buffer. The proteins bound to the immobilized DNA were boiled for 2 min at 95 °C in an oxidative SDS-loading buffer and analyzed by SDS-PAGE using SuperSep$^{TM}$ Ace, 5–20% gel (Wako, Japan). At least three independent experiments were performed.

## *Xenopus* egg extracts

*Xenopus laevis* was purchased from Kato-S Kagaku and handled according to the animal care regulations at the University of Tokyo. The preparation of interphase egg extracts, chromatin isolations, and immunodepletions were performed as described previously[24,40]. Unfertilized *Xenopus laevis* eggs were dejellied in 2.5% thioglycolic acid-NaOH [pH 8.2] and washed in 1xMMR buffer (100 mM NaCl, 2 mM KCl, 1 mM MgCl$_2$, 2 mM CaCl$_2$, 0.1 mM EDTA, and 5 mM HEPES-NaOH [pH 7.5]). After activation in 1xMMR supplemented with 0.3 µg/mL calcium ionophore, eggs were washed with EB buffer (50 mM KCl, 2.5 mM MgCl$_2$, 10 mM HEPES-KOH [pH 7.5], 50 mM sucrose). Eggs were packed into tubes by centrifugation (BECKMAN, Avanti J-E, JS-13.1 swinging rotor) for 1 min at 190×$g$ and crushed by centrifugation for 20 min at 18,973×$g$. Egg extracts were supplemented with 50 µg/mL cycloheximide, 20 µg/mL cytochalasin B, 1 mM DTT, 2 µg/mL aprotinin, 5 µg/mL leupeptin, and clarified by ultracentrifugation (Hitachi, CP100NX, P55ST2 swinging rotor) for 20 min at 48,400×$g$. The cytoplasmic extracts were aliquoted, frozen in liquid nitrogen, and stored at −80 °C. All extracts were supplemented with an energy regeneration system (2 mM ATP, 20 mM phosphocreatine, and 5 µg/ml creatine phosphokinase). About 3000–4000 nuclei/µl of sperm nuclei were added and incubated at 22 °C. Aliquots (15–20 µl) were diluted with 150 µl chromatin purification buffer (CPB; 50 mM KCl, 5 mM MgCl$_2$, 20 mM HEPES-KOH [pH 7.6]) containing 0.1% NP-40, 2% sucrose, 2 mM *N*-ethylmaleimide (NEM). After incubation on ice for 5 min, diluted extracts were layered over 1.5 mL of CPB containing 30% sucrose and centrifuged at 15,000×$g$ for 10 min at 4 °C. Chromatin pellets were resuspended in 1×Laemmli sample buffer, boiled for 5 min at 100 °C, and analyzed by immunoblotting using Rabbit anti-Xenopus PAF15 antibody (produced and validated by Nakanishi lab, the University of Tokyo, 1:500 dilution for WB), Rabbit anti-Xenopus DNMT1 (produced and validated by Nakanishi lab, the University of Tokyo, 1:500 dilution for WB), Rabbit anti-Xenopus UHRF1 (1:500 dilution for WB) (produced and validated by Nakanishi lab, the University of Tokyo), Rabbit anti-Xenopus ORC2 (validated and provided by J. Maller, University of Colorado, 1:500 dilution for WB), Rabbit anti-USP7 (A300-033A; Bethyl, 1:100 dilution for WB), Rabbit anti-histone H3 (ab1791; Abcam, 1:3000 dilution for WB) and Mouse anti-PCNA (PC10) (sc-56; Santa Cruz Biotechnology, 1:1000 dilution for WB).

For protein expression in insect cells, baculoviruses were produced using a BD BaculoGold Transfection kit and a BestBac Transfection kit (BD Biosciences), following the manufacturer's protocol. Proteins were expressed in Sf9 insect cells by infection with viruses expressing xDNMT1 Wt-FLAGx3, xDNMT1 F506AF507A-FLAGx3 for 72 h. Sf9 cells from a 500 ml culture were collected and lysed by resuspending in 20 mL lysis buffer (20 mM Tris·HCl [pH 8.0], 100 mM KCl, 5 mM MgCl$_2$, 10% glycerol, 1% NP-40, 1 mM DTT, 10 µg/mL leupeptin, and 10 µg/mL aprotinin), followed by incubation on ice for 10 min. A soluble fraction was obtained after centrifugation of the lysate at 15,000×$g$ for 15 min at 4 ˚C. The soluble fraction was incubated for 4 h at 4 ˚C with 250 µL of anti-FLAG M2 affinity resin (Sigma-Aldrich) equilibrated with lysis buffer. The beads were collected and washed with 10 mL wash buffer (20 mM Tris·HCl [pH 8.0], 100 mM KCl, 5 mM MgCl$_2$, 10% glycerol, 0.1% NP-40, 1 mM DTT) and then with 5 mL EB (20 mM HEPES-KOH [pH 7.5], 100 mM KCl, 5 mM MgCl$_2$) containing 1 mM DTT. The recombinant xDNMT1 was eluted twice in 250 µL EB containing 1 mM DTT and 250 µg/mL 3×FLAG peptide (Sigma-Aldrich). Eluates were pooled and concentrated using a Vivaspin 500 (GE

Healthcare Biosciences). F506A/F507A mutations of xDNMT1 were generated using two primers containing the mutations (Forward: 5′-GCCGCCTCCGAACAAATCGAAAAGGATGCAG-3′, Reverse: 5′- TGTGT CAAAAATCTGATACACAAGT-3′).

## Cell culture, transfection, and colony isolation

The HCT116 cell line, which conditionally expressed OsTIR1 under the control of a tetracycline (Tet)-inducible promoter, was obtained from the RIKEN BRC Cell Bank (http://cell.brc.riken.jp/en/), and genotyped by Eurofins. All cell lines were cultured in McCoy's 5 A medium (Sigma-Aldrich) supplemented with 10% FBS (Gibco), 2 mM L-glutamine, 100 U/mL penicillin, and 100 μg/mL streptomycin. Cells were grown in a 37 °C humid incubator with 5% $CO_2$. To generate stable DNMT1-AID cell lines, we followed previous studies[41,42]. Briefly, cells were grown in a 24-well plate, then CRISPR/Cas and donor plasmids were transfected using Lipofectamine 2000 (Thermo Fisher Scientific). Two days after transfection, cells were transferred and diluted in 10 cm dishes, followed by selection in the presence of 700 mg/mL G418 or 100 mg/mL Hygromycin B. After 10–12 days, colonies were picked for further selection in a 96-well plate. To induce the degradation of AID-fused proteins, cells were incubated with 0.2 μg/mL doxycycline (Dox) and 20 μM auxinole for 1 day, then we replaced the medium including 0.2 μg/mL Dox and 500 μM indole-3-acetic acid (IAA), a natural auxin.

## Establishment of stable expressing exogenous DNMT1 cell lines for rescue experiments

We cloned WT DNMT1 and point mutant DNMT1 (F631A/F632A) to pSBbi-Bla (Addgene: 60526). All plasmids were sequenced prior to use. To establish stable expressing exogenous DNMT1 cell lines, we used the sleeping beauty system[43]. Briefly, cells were grown in a six-well plate, then transposase vector (Addgene: 34879) and each pSBbi-Bla plasmids (EV, WT, F631A/F632A) were transfected using Lipofectamine 2000 (Thermo Fisher Scientific). Four days after transfection, cells were selected with 10 μg/mL Blasticidin for 1 week. To detect DNA methylation levels, LUMA and Pyrosequencing were done according to standard procedures. Depletion of the endogenous DNMT1 was confirmed by immunoblotting using rabbit anti-DNMT1 (#5032; CST, 1:1000 dilution), mouse anti-Tubulin (ab7291; Abcam, 1:4000 dilution), and rabbit anti-V5 (ab206566; Abcam, 1:100 dilution). F631A/F632A site-directed mutagenesis of DNMT1 was performed by inverse PCR using two primers containing the mutations (Forward: 5′- ATACTG CCGCCGCAGAGCAAATTGAAAAGGATG-3′, Reverse: 5′-TCTGCGGCGG CAGTATCGAAGATCTGGTAGACC-3′).

## Reporting summary

Further information on research design is available in the Nature Portfolio Reporting Summary linked to this article.

# Data availability

The data that support this study are available from the corresponding author upon reasonable request. The cryo-EM density map has been deposited in the Electron Microscopy Data Bank (EMDB, www.ebi.ac.uk/pdbe/emdb/) under accession code EMD-33200, EMD-33201, EMD-33298, EMD-33299, and the atomic coordinates of CXXC-ordered and CXXC-disordered ternary complex have been deposited in the PDB (www.rcsb.org) under accession code 7XI9 and 7XIB, respectively. All data needed to evaluate the conclusions in the paper are present in the paper and/or the Supplementary Materials. PDB 4WXX, 3PTA, 6X9I, 4DA4, and 5WVO were used for this study. Source data are provided with this paper.

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

## Acknowledgements

The cryo-EM experiments were performed at the cryo-EM facility of the RIKEN Center for Biosystems Dynamics Research in Yokohama. We thank the members of the Structure Biology Research Center at KEK, especially M. Kawasaki and T. Senda for cryo-EM and N. Shimizu for SAXS. We thank the members of the cryo-EM facility at the University of Tokyo, especially Y. Sakamaki and M. Kikkawa. M. Ariyoshi supports the preparation of the manuscript. This work was supported by JSPS KAKENHI under Grant numbers 18H02392, 19H05294, 19H05741 (K.A.), 19J22030 (S.K.), 19H05285, 21H00272 (A.N.), the grant for 2021–2023 Strategic Research Promotion (No. SK201904) of Yokohama City University (K.A.), Sasakawa Scientific Research Grant 2171000003 (H.O.). P.A.D. is supported by Agence Nationale de la Recherche (PRCI INTEGER ANR-19-CE12-0030-01), LabEx "Who Am I?" (ANR-11-LABX-0071), Université de Paris IdEx (ANR-18-IDEX-0001) funded by the French Government through its "Investments for the Future" program, Fondation pour la Recherche Médicale, Fondation ARC (Program Labellisé PGA1/RF20180206807). K.Y. was the recipient of a postdoctoral fellowship from the Fondation Association pour la Recherche sur le Cancer, and of a subsequent postdoc fellowship from Labex "Who Am I?". This research was partially supported by Platform Project for Supporting Drug Discovery and Life Science Research (Basis for Supporting Innovative Drug Discovery and Life Science Research [BINDS]) from AMED under Grant Number 1770, 2102, 2965, and 3002.

## Author contributions

K.A. supervised the work. A.K., H.O., S.K., S.M., S.Y., and H.S. performed cloning and protein purification. A.K., H.O., S.Y., and H.S. performed in vitro biochemical assay. S.K. and K.A. performed SEC-SAXS experiments and analysis. A.K., H.O., A.Y., M.S., N.A., and K.A. performed Cryo-EM experiments and analysis. Y.C., S.T., M.N., and A.N. performed the evaluation of mutant using *Xenopus* egg extract and K.Y., J.S., H.K., and P.A.D. performed cell-based assay. A.K., H.O., P.A.D., and K.A. wrote the paper.

## Competing interests

The authors declare no competing interests.
