## [Peer Review File · Nature Communications]

Reviewers' Comments:

Reviewer #1:

Remarks to the Author:

This study reported the cryo-EM structure of nearly full-length human DNMT1 bound to its two natural activators: hemimethylated DNA and ubiquitinated histone H3. They showed an unstudied linker, between the RFTS and CXXC domains, plays a key role for activation. Altogether this study provides a new model for activation of DNMT1 for DNA maintenance methylation and is of broad interest. A few issues need to be addressed or discussed before I will support for publication.

1. Authors should explain why a truncated DNMT1 (aa:351-1616) but not the full-length protein was used for the study.
2. In a recent paper published by the same groups (Nat Commun. 2020 Mar 6;11(1):1222. doi: 10.1038/s41467-020-15006-4), PAF15Ub2 was proposed to serve as the primary with H3Ub2 as a backup for DNMT1 recruitment and activation. It would be important to test or compare if PAF15Ub2 activates DNMT1 in a similar way.
3. In Figure 4C, authors marked H3 and ubiquitinated H3 with red box. It would be helpful to readers if different forms of ubiquitinated H3 were indicated.
4. As shown in Figure 4C and especially in mammalian cells in other published studies, mono-ubiquitinated H3 appeared to be dominant, raising the question whether mono-ubiquitinated H3 could activate DNMT1. In Figure 4b, authors used a nice methylation assay to compare the activity of the wildtype and F631A/F632A mutant. Thus, a similar experiment could be performed to determine if activation of DNMT1 is indeed dependent on H3Ub2 and could not be activated by the more common H3Ub1.

Reviewer #2:

Remarks to the Author:

DNMT1 is an essential enzyme that maintains genomic DNA methylation. As a multiple domain-containing protein, DNMT1 is subject to intramolecular regulations that strongly restrict its activity to hemimethylated DNA. To understand the detailed molecular mechanism for DNMT1 activation, in this study, the authors determined the cryo-EM structure of nearly full-length human DNMT1 (aa.351-1616), stimulated by the H3Ub2 tail and in an intermediate complex with a hemimethylated DNA analog. The structure illuminates the synergistic structural rearrangements that underpin the activation of DNMT1, and also highlights the key role of a hydrophobic "Toggle" pocket in the catalytic domain. In addition, the authors identified a hitherto unstudied linker between the RFTS and CXXC domains, which plays a key role in activation. Overall, the data and interpretation are reasonable, and the structural work makes a meaningful contribution to our understanding of the mechanistic basis for the activation of DNMT1. However, I have the following concerns that the authors should address before the publication of the manuscript.

1. The Paragraph 2 on page 7, to separate the contributions of H3Ub2 and hemimethylated DNA for the displacement of the RFTS, the authors determined the cryo-EM structure of DNMT1 (aa:351-1616) in complex with H3Ub2, but without DNA. The authors found that H3Ub2-binding to the RFTS domain might not be sufficient for displacement of the RFTS domain. However, the authors draw the conclusion that upon joint binding of H3Ub2 and hemimethylated DNA, the RFTS domain is forced out of the catalytic core. The role of hemimethylated DNA in the displacement of the RFTS domain from the catalytic domain is not clear in this manuscript. It is my understanding that the binary complex of DNMT1 (aa:351-1616) bound to hemimethylated DNA might be required to provide some information.
2. The authors propose that the molecular mechanism for human DNMT1 catalytic activation may be limited to animals, as the key phenylalanine of the Activating Helix is missing from other DNA methyltransferases, such as plant DNA methyltransferases maize CMT3 homolog ZMET2 and *Nicotiana tabacum* DRM2. In the mutagenesis section, the authors only demonstrated the results for F631A/F632A mutations that abolished the ability of DNMT1 to bind hemimethylated DNA. Are the corresponding residues in those plant DNA methyltransferases alanine? I wonder whether

other residues can regulate the enzymatic activity. Without this data, the authors have not sufficiently addressed their hypothesis.

3. Minor comment: The authors point out that the "Toggle" pocket is composed of Val1248, Phe1263, 152 Leu1265, Phe1274, Val1279, and Leu1282, I would suggest that the authors label the residues in figures to improve its readability.

4. The paragraph 4 on page 8, the authors point out that "in the inactive state, the DNA Recognition Helix (aa:1236-1259) in the catalytic domain is kinked, at Ser1245", Ser1245 should be labeled in figure 3b or 3c.

5. The paragraph 3 on page 8, "Zooming in on the catalytic site, we observed another striking difference between the inactive and active states of the enzyme (Fig. 3a)." please verify the figure citation.

Reviewer #3:

Remarks to the Author:

This paper reports the structure of the core domain of DNMT1 determined by cryoEM in the presence of hemi methylated DNA and ubiquitinated histone H3. Although they included nearly full length human DNMT1 in the sample preparation they were only able to determine the core of the structure (amino acids 614 -1609). The structure of this region of the protein has been reported previously in several crystal structures. However, this report does represent a significant advance since the structure of this core domain is different in terms of a region termed "activating helix" that appears to adopt a different conformation in this structure. The authors propose that this is a result of the RFTS domain binding to a ubiquitinated peptide H3Ub2. This is significant because this activating helix displaces an inhibitory linker and results in reorientating the DNA recognition helix and hence activating the enzyme. This finding of a mechanism for DNMT1 activation represents an advance that is potentially suitable for publication in Nature Communications. However, there are a number of concerns that need to be addressed.

First the abstract of this manuscript claims to report the structure of nearly full length human DNMT1. However since the PDB entries do not contain the histone H3Ub2 peptide and much of the N-terminal region of DNMT1 the abstract of the paper is seriously misleading. Indeed, the manuscript needs to be much clearer on how it expands on existing knowledge since there are many existing structures of DNMT1: 3PTA DNMT1 (646-1600) with DNA; 4WXX DNMT1 (351-1605); 6X9I DNMT1 (729-1600) with DNA; 6PZV DNMT1 (351-590); 5WVO DNMT1 (365-591) with H3Ub2 peptide. Figure 1 presents a segmented EM map. It would be much better if it included a fitted structure along with the map.

Second, the structural rearrangements described in figure 2 are unclear since the figure shows a combination of segmented map, cartoon and boxed enlargements. It is impossible to make direct comparisons between the existing structures and the new structures.

In the main text Table 1 reports two structures with both with PDB and EMDB entries. The two structures (7X19 and 7X1B), contain amino acids 614 -1609 of DNMT1 with 12 base pairs of DNA. There are no EMDB validation reports for the table 1 structures.

Supplementary table 1 reports two structures both with EMDB entries.

The supplementary table 1 reports EM maps for "human DNMT1 351-1616" and "human DNMT1 351-1616 in the presence of Ubiquitinated H3Ub2". There are EMDB validation reports for these maps but not associated PDB entry. Hence the implication that the longer structure has been solved is misleading.

Overall, it remains unclear how H3Ub2 binding to the RFTS domain results in the change in conformation of the activation helix and hence activation of the enzyme. This needs to be discussed in more detail.

Reviewer #1 (Remarks to the Author):

This study reported the cryo-EM structure of nearly full-length human DNMT1 bound to its two natural activators: hemimethylated DNA and ubiquitinated histone H3. They showed an unstudied linker, between the RFTS and CXXC domains, plays a key role for activation. Altogether this study provides a new model for activation of DNMT1 for DNA maintenance methylation and is of broad interest.

> We thank the reviewer for expressing their interest in our work.

A few issues need to be addressed or discussed before I will support for publication.

1. Authors should explain why a truncated DNMT1 (aa:351-1616) but not the full-length protein was used for the study.

[Our response] Thank you for this question. We also tried to prepare full-length DNMT1 from Sf9 cells, but our best efforts have been unsuccessful, as the protein underwent degradation during the purification. The truncated DNMT1 fragment we have worked with (aa:351-1616) is the largest we could produce. We have added following sentence in the revised manuscript.

Page 5, line 15

'The human DNMT1 protein, **a minimum fragment required for investigating the activation mechanism by binding of ubiquitinated H3**, was produced using the Sf9 baculovirus expression system'.

2. In a recent paper published by the same groups (Nat Commun. 2020 Mar 6;11(1):1222. doi: 10.1038/s41467-020-15006-4), PAF15Ub2 was proposed to serve as the primary with H3Ub2 as a backup for DNMT1 recruitment and activation. It would be important to test or compare if PAF15Ub2 activates DNMT1 in a similar way.

[Our response] This point is important and thus well taken. We have prepared PAF15Ub2, and performed an *in vitro* DNA methylation assay. As you can see below, the double-monoubiquitinated PAF15 enhanced the DNA methylation activity of DNMT1 in a manner similar to ubiquitinated H3, which answers your question positively.

However, we respectfully feel that is best to leave this data out of the current paper. Indeed, the current paper is focused on regulation of DNMT1 by H3 (and not PAF15), and we are currently working on a future manuscript specifically dedicated to the regulation of DNMT1 by PAF15. We feel this piece of data belongs in this future manuscript.

3. In Figure 4C, authors marked H3 and ubiquitinated H3 with red box. It would be helpful to readers if different forms of ubiquitinated H3 were indicated.

[Our response] Thank you for this suggestion. We have added the labels of H3, H3Ub1 and H3Ub2 in the revised Fig. 4c.

4. As shown in Figure 4C and especially in mammalian cells in other published studies, mono-ubiquitinated H3 appeared to be dominant, raising the question whether mono-ubiquitinated H3 could activate DNMT1. In Figure 4b, authors used a nice methylation assay to compare the activity of the wildtype and F631A/F632A mutant. Thus, a similar experiment could be performed to determine if activation of DNMT1 is indeed dependent on H3Ub2 and could not be activated by the more common H3Ub1.

[Our response] Thank you for this question. In order to answer it, we have prepared K18 or K23 single single-monoubiquitinated H3 peptide and conducted *in vitro* DNA methylation assay. As predicted from the previous reports (Ishiyama, Mol Cell, 2017; Mishima, Genes to

Cells, 2022), the single-monoubiquitinated H3 enhanced the enzymatic activity of DNMT1, but at a lower efficiency than double-monoubiquitinated H3, which indicate that double-monoubiquitinated H3 functions as a specific signal for enzymatic activation of DNMT1. We have replaced the data about *in vitro* methylation assay in Extended Data Fig.1e with new one and added the following text in the revised manuscript.

Page 5, line 19

'As expected from previous work^{12,13}, in contrast to K18 or K23 single monoubiquitinated H3, the addition of H3Ub2 effectively enhanced the enzymatic activity of DNMT1 (Extended Data Fig. 1d,e).'

Reviewer #2 (Remarks to the Author):

DNMT1 is an essential enzyme that maintains genomic DNA methylation. As a multiple domain-containing protein, DNMT1 is subject to intramolecular regulations that strongly restrict its activity to hemimethylated DNA. To understand the detailed molecular mechanism for DNMT1 activation, in this study, the authors determined the cryo-EM structure of nearly full-length human DNMT1 (aa.351-1616), stimulated by the H3Ub2 tail and in an intermediate complex with a hemimethylated DNA analog. The structure illuminates the synergistic structural rearrangements that underpin the activation of DNMT1, and also highlights the key role of a hydrophobic "Toggle" pocket in the catalytic domain. In addition, the authors identified a hitherto unstudied linker between the RFTS and CXXC domains, which plays a key role in activation. Overall, the data and interpretation are reasonable, and the structural work makes a meaningful contribution to our understanding of the mechanistic basis for the

activation of DNMT1.

> We thank you for expressing interest in our work

However, I have the following concerns that the authors should address before the publication of the manuscript.

1. The Paragraph 2 on page 7, to separate the contributions of H3Ub2 and hemimethylated DNA for the displacement of the RFTS, the authors determined the cryo-EM structure of DNMT1 (aa:351-1616) in complex with H3Ub2, but without DNA. The authors found that H3Ub2-binding to the RFTS domain might not be sufficient for displacement of the RFTS domain. However, the authors draw the conclusion that upon joint binding of H3Ub2 and hemimethylated DNA, the RFTS domain is forced out of the catalytic core. The role of hemimethylated DNA in the displacement of the RFTS domain from the catalytic domain is not clear in this manuscript. It is my understanding that the binary complex of DNMT1 (aa:351-1616) bound to hemimethylated DNA might be required to provide some information.

[Our response] This point is important and thus well taken. We tried to prepare the DNMT1:DNA (containing 5-fluorocytosine) binary complex and analyze the complex formation by gel filtration chromatography. As shown in the Extended data Fig. 1c, DNMT1 failed to form the binary complex with the DNA, which means that RFTS domain is not released from the catalytic core. Thus, we concluded that simultaneous binding of ubiquitinated H3 and hemimethylated DNA is required for the enzymatic activation of DNMT1. This data is displayed in Extended Data Fig. 1c, and we have added the following text in the revised manuscript.

Page 12, line 22.

'In addition, apo-DNMT1 was unable to form the binary complex with hemimethylated DNA (Extended Data Fig. 1C)'

2. The authors propose that the molecular mechanism for human DNMT1 catalytic activation may be limited to animals, as the key phenylalanine of the Activating Helix is missing from other DNA methyltransferases, such as plant DNA methyltransferases maize CMT3 homolog ZMET2 and Nicotiana tabacum DRM2. In the mutagenesis section, the authors only demonstrated the results for F631A/F632A mutations that abolished the ability of DNMT1 to bind hemimethylated DNA. Are the corresponding residues in those plant DNA methyltransferases alanine? I wonder whether other residues can regulate the enzymatic

activity. Without this data, the authors have not sufficiently addressed their hypothesis.

[Our response] Thank you for this insightful note. As suggested, we cannot experimentally demonstrate whether other residues regulate the DNA methylation activity in plant methyltransferase. We think the discussion in the previous manuscript was speculative. Therefore, we have removed the corresponding text in the revised manuscript.

3. Minor comment: The authors point out that the "Toggle" pocket is composed of Val1248, Phe1263, 152 Leu1265, Phe1274, Val1279, and Leu1282, I would suggest that the authors label the residues in figures to improve its readability.

[Our response] Thank you for this comment. We have amended Fig. 3b and 3c.

4. The paragraph 4 on page 8, the authors point out that "in the inactive state, the DNA Recognition Helix (aa:1236-1259) in the catalytic domain is kinked, at Ser1245", Ser1245 should be labeled in figure 3b or 3c.

[Our response] Thank you for the comment. We have added the label of Ser1246 in Fig 3b and 3c.

5. The paragraph 3 on page 8, "Zooming in on the catalytic site, we observed another striking difference between the inactive and active states of the enzyme (Fig. 3a)." please verify the figure citation.

[Our response] Thank you for pointing this out. We have removed incorrect figure citation.

Reviewer #3 (Remarks to the Author):

This paper reports the structure of the core domain of DNMT1 determined by cryoEM in the presence of hemi methylated DNA and ubiquitinated histone H3. Although they included nearly full length human DNMT1 in the sample preparation they were only able to determine the core of the structure (amino acids 614 -1609). The structure of this region of the protein has been reported previously in several crystal structures. However, this report does represent a significant advance since the structure of this core domain is different in terms of a region termed "activating helix" that appears to adopt a different conformation in this

structure. The authors propose that this is a result of the RFTS domain binding to a ubiquitinated peptide H3Ub2. This is significant because this activating helix displaces an inhibitory linker and results in reorientating the DNA recognition helix and hence activating the enzyme. This finding of a mechanism for DNMT1 activation represents an advance that is potentially suitable for publication in Nature Communications.

> We thank the reviewer for their support

However, there are a number of concerns that need to be addressed.

First the abstract of this manuscript claims to report the structure of nearly full length human DNMT1. However since the PDB entries do not contain the histone H3Ub2 peptide and much of the N-terminal region of DNMT1 the abstract of the paper is seriously misleading. Indeed, the manuscript needs to be much clearer on how it expands on existing knowledge since there are many existing structures of DNMT1: 3PTA DNMT1 (646-1600) with DNA; 4WXX DNMT1 (351-1605); 6X9I DNMT1 (729-1600) with DNA; 6PZV DNMT1 (351-590); 5WVO DNMT1 (365-591) with H3Ub2 peptide. Figure 1 presents a segmented EM map. It would be much better if it included a fitted structure along with the map.

[Our response] Thank you for the comments. We agree with your suggestion. We have removed the 'nearly full-length' in the revised manuscript and amended the Figs 1a and 1b according to the reviewer's suggestion. We have amended the manuscript as follows.

Page 4, line 14

'In the absence of DNA (apo-DNMT1, PDB:4WXX, aa:351-1600), the enzyme is autoinhibited: Binding of Replication-Foci Targeting Sequence (RFTS) to the catalytic core, in association with recognition of the DNA binding region by an Auto-Inhibitory Linker, inhibit the access of hemimethylated DNA to DNMT1 catalytic region 16,17. A key unresolved question is: how does the combined presence of H3Ub2 and hemimethylated DNA allow the enzyme to overcome this double inhibition? Of note, previous structural studies of DNMT1 in a complex with hemimethylated DNA (PDB:4DA4, aa:731-1602; 6X9I, aa:729-1600) 18–20, in a complex with unmethylated CpG DNA (PDB:3PTA, aa:646-1600) 21 and RFTS bound to H3Ub2 (PDB:5WVO, aa:351-600; 6PZV, aa:349-594) 12,13 have used a truncated version of the protein (Fig. 1a), therefore the fate of the inhibitory regions, RFTS and Auto-Inhibitory Linker, during activation is unknown.'

Second, the structural rearrangements described in figure 2 are unclear since the figure

shows a combination of segmented map, cartoon and boxed enlargements. It is impossible to make direct comparisons between the existing structures and the new structures.

[Our response] Thank you for helping us improve the figure. We have amended Fig. 2a to show the structural differences clearly.

In the main text Table 1 reports two structures with both with PDB and EMDB entries. The two structures (7X19 and 7X1B), contain amino acids 614 -1609 of DNMT1 with 12 base pairs of DNA. There are no EMDB validation reports for the table 1 structures.

[Our response] This issue has also been pointed out by the editor. We apologize for forgetting to join the validation reports upon first submission. We have submitted the validation reports this time.

Supplementary table 1 reports two structures both with EMDB entries.

The supplementary table 1 reports EM maps for “human DNMT1 351-1616” and “human DNMT1 351-1616 in the presence of Ubiquitinated H3Ub2”. There are EMDB validation reports for these maps but not associated PDB entry. Hence the implication that the longer structure has been solved is misleading.

[Our response] Thank you for the comment. As described in the method section, we did not construct the atomic model of human apo-DNMT1 351-1616 and human apo-DNMT1 351-1616 in the presence of H3Ub2 because of low resolution of the cryo-EM map. However, we have submitted the cryo-EM map to EMDB.

Overall, it remains unclear how H3Ub2 binding to the RFTS domain results in the change in conformation of the activation helix and hence activation of the enzyme. This needs to be discussed in more detail.

[Our response] This point is very important and thus well taken. As described in the criticism 1 from reviewer 2, we tried to prepare DNMT1 in complex with hemimethylated DNA, but failed to do so. This data strengthens that simultaneous binding of H3Ub2 and hemimethylated DNA is required for the enzymatic activation of DNMT1. However, in this study, we could not reveal the direct evidence underlying the conformational change of Activating Helix upon binding of H3Ub2. We have added this point in a consideration for future study. We have added this point into Discussion in page 13, line 7.

'However, it is currently unknown how simultaneous binding of H3Ub2 and hemimethylated DNA causes a conformational change in the Activation Helix to place F631/F631 in the Toggle Pocket. Future work, such as molecular dynamics simulation, will determine if these structural changes occur sequentially or simultaneously.'

Reviewers' Comments:

Reviewer #1:

Remarks to the Author:

The authors have addressed all my concern and I have no further question.

Reviewer #2:

Remarks to the Author:

The concerns have been addressed in the revisions. This finding of a mechanism for DNMT1 activation is suitable for publication in Nature Communications.

Reviewer #3:

Remarks to the Author:

I am pleased to say that my concerns have been addressed.

The manuscript and figures have been suitably modified.

The comparisons with previous structures now clearly show the conformational changes of the inhibitory loop.